# ProteinWeaver: A webtool to visualize ontology-annotated protein networks

**Oliver Anderson**[ID]☯, **Altaf Barelvi**☯, **Aden O'Brien**, **Ainsley Norman**, **Iris Jan**, **Anna Ritz**[ID]*

Biology Department, Reed College, Portland, Oregon, United States of America

☯ These authors contributed equally to this work.
* aritz@reed.edu

**Data availability statement:** ProteinWeaver is licensed under GNU General Public License v3.0 and is available at https://proteinweaver.reedcompbio.org/. All data

## Abstract

Molecular interaction networks are a vital tool for studying biological systems. While many tools exist that visualize a protein or a pathway within a network, no tool provides the ability for a researcher to consider a protein's position in a network in the context of a specific biological process or pathway. We developed ProteinWeaver, a web-based tool designed to visualize and analyze non-human protein interaction networks by integrating known biological functions. ProteinWeaver provides users with an intuitive interface to situate a user-specified protein in a user-provided biological context (as a Gene Ontology term) in seven model organisms. ProteinWeaver also reports the presence of physical and regulatory network motifs within the queried subnetwork and statistics about the protein's distance to the biological process or pathway within the network. These insights can help researchers generate testable hypotheses about the protein's potential role in the process or pathway under study. Two cell biology case studies demonstrate Protein-Weaver's potential to generate hypotheses from the queried subnetworks. ProteinWeaver is available at https://proteinweaver.reedcompbio.org/.

## Introduction

Biological networks are essential tools for modeling and understanding complex biological systems [1,2]. Protein-protein interaction (PPI) networks are biological networks that describe all the proteins and their molecular interactions within a biological system [3]. Species-specific PPI networks were first built from high-throughput protein interaction detection methods (such as yeast two-hybrid (Y2H) and affinity purification mass spectrometry (AP/MS) studies) [3] and are now generated from a diverse array of *in vitro*, *in vivo*, and *in silico* methods [4]. Graphs are powerful tools for modeling and analyzing the function and interactions of proteins within an organism [5,6], and have been used to identify the differences between healthy and disease states in organisms [7]. PPI networks are valuable tools for analyzing molecular interactions across all domains of life, from viral to human tissue, and PPIs from non-human model organisms can be used to study human diseases [8].

In addition to PPI networks, gene regulatory networks (GRNs) are graphical representations of biological systems utilized for studying development, differences between healthy

have been deposited to Zenodo at https://zenodo.org/records/15843724. Data, scripts, and website source code are available at https://github.com/Reed-CompBio/protein-weaver/. Protein Prediction data is available at https://github.com/Reed-CompBio/protein-function-prediction.

**Funding:** This work was supported by a National Science Foundation (https://www.nsf.gov/) grant NSF-DBI-1750981 awarded to AR. The funders not play any role in the study design, data collection and analysis, decision to publish, or preparation of the manuscript.

**Competing interests:** The authors have declared that no competing interests exist.

and diseased states, and other biological processes that are regulated by transcription factors (TFs) [9]. GRNs differ slightly from PPI networks in their graphical representation: there are directed interactions where TFs regulate a gene or gene product rather than undirected physical interactions. Although many existing tools represent PPI networks or GRNs graphically, many represent one type of network in isolation or fail to draw clear visual distinctions between the interaction types [10]. In addition, regulatory interactions and PPIs do not exist in isolation; physical and gene regulatory interactions coexist, forming intricate patterns or motifs. Physical interactions in these "mixed motifs" have been found to participate in regulatory interactions [11]. Methods to identify mixed motifs are lacking but can be useful in discovering clusters of proteins or gene products for further experimental analysis [10].

Various web tools have been developed to visualize molecular networks. STRING-DB is one of the most well-known network visualization tools, offering the ability to visualize subnetworks in different species by querying proteins or biological processes [12]. STRING-DB offers a variety of features, such as searching for protein information and displaying direct neighbors through a protein interaction network. One of its recent additions allows users to search for proteins associated with specific pathways. Another network visualization tool, GeneMANIA [13], uses additional functional genomic data to generate subnetworks. GeneMANIA is a web-based tool for visualizing biological networks. Given a specific gene, it generates a network visualization that connects the gene to its first-degree neighbors, showing various types of edges, such as physical interactions and co-expression. Users can adjust the number of neighboring genes displayed, and information on both proteins and edges is accessible through interactions with the visualization. However, both the tools lack the ability to fully explore the connections between a protein and specific pathways or biological processes. The pathway feature in STRING-DB is limited to pathways with fewer than 2,000 proteins and generates dense visualizations, and GeneMANIA does not allow users to query based on biological function. Further, while both tools offer network visualization parameters, they do not encourage interactively traversing the broader network.

In addition to STRING-DB and GeneMANIA, additional visualization tools are focused on specific subsets of the larger networks, such as signaling pathway visualization tools like SignaLink [14] and KEGG [15]. Many visualization tools are tailored for individual species, such as BioPlex [16] and GenePlexus [17] for humans, the *Drosophila* Interactions Database [18] for flies, and SubtiWiki for *B. subtilis* [19]. Finally, other network visualization tools such as KeyPathwayMiner [20] and NetworkAnalyst [21] require user data as input and/or run specialized algorithms. Although the network visualization capabilities of these tools are comprehensive, none of these existing tools allow users to search by a protein and specific pathway and get a subnetwork grouped by both their function and interactions with the protein of interest. Additionally, some tools that run algorithms hide the details from the user, so it is unclear to the researcher how the subnetworks are selected for visualization.

To address these challenges with existing tools, we developed ProteinWeaver, a molecular interaction network visualization tool that generates subnetworks of physical and regulatory interactions based on a protein and a biological function of interest for non-human model organisms. In contrast to previous tools, ProteinWeaver links proteins to relevant biological processes, provides customizable network visualizations, and encourages interactive network exploration. To our knowledge, ProteinWeaver is the first interactive webtool to combine physical and regulatory interactions within a single network. Currently, ProteinWeaver supports two prokaryotes (the Gram-positive bacterium *Bacillus subtilis subsp. subtilis str. 168* and the Gram-negative bacterium *Escherichia coli K-12*), a single-celled eukaryote (the brewer's yeast *Saccharomyces cerevisiae S288C*), two morphologically-distinct invertebrates (the fruit fly *Drosophila melanogaster* and the nematode *Caenorhabditis elegans*), a vertebrate

(the zebrafish *Danio rerio*), and a flowering plant (*Arabidopsis thaliana*). ProteinWeaver characterizes biological functions with the Gene Ontology (GO), a collection of classifications for protein function (GO terms) [22,23]. GO terms are a valuable tool for predicting protein function using PPI networks and can help situate a protein within a biological context [24]. ProteinWeaver generates a subnetwork that connects a protein of interest with proteins annotated to the specified biological context. The graphical interface is fast, visually intuitive, and does not require previous computational experience to use effectively.

ProteinWeaver offers two additional pieces of information to help situate a protein in the context of a biological process or pathway. It counts five different network motifs (one with PPI edges, another with regulatory edges, and three with a mix of PPI and regulatory edges) and provides enrichment scores for users to understand the expected motif count for an organism. ProteinWeaver also provides a quantitative measure that describes how close a protein is to proteins annotated to a biological process within the network. These additional features help provide context for the protein of interest and the surrounding molecular interactions for hypothesis generation.

## Materials and methods

### Interaction data

Currently, ProteinWeaver supports physical and regulatory interaction data for seven non-human model organisms (Table 1). For each organism, we collected experimental, text-mined, and database-validated protein and genetic interactions, as well as GO annotations (Sect A1.1 in S1 File). All protein-protein and regulatory interactions are linked to evidence sources when available, such as PubMed or STRING-DB references [12].

The relationships among proteins and GO terms are represented as a connected graph with two types of nodes (proteins and GO terms) and three types of edges that capture undirected physical interactions among proteins, directed regulatory interactions among proteins, and directed GO term annotations between GO terms and proteins (Fig 1A and Sect A1.2 in S1 File). We add the directly annotated protein-GO Term pairs into the graph and infer annotations between proteins and more general GO terms (dashed blue edges in Fig 1A). ProteinWeaver uses the Neo4j Graph Database to store and query the graph. See Sect A1.3 in S1 File for more information about Neo4j and Sect A1.4 in S1 File for information about the ProteinWeaver development stack.

### Queried subnetworks

The fundamental feature of ProteinWeaver is the ability for a user to enter a query protein *s* and a GO term *t* and visualize the connections from *s* to proteins annotated to *t*. We use the

**Table 1**. **Summary of ProteinWeaver's underlying network data.** Nodes represent proteins and their encoding genes. PPI: protein-protein interaction. GRN: gene regulatory network. GO: Gene Ontology.

| Organism | Nodes | PPI Edges | GRN Edges | GO Terms | GO Annotations |
|---|---|---|---|---|---|
| *A. thaliana* | 8,583 | 51,384 | 1,375 | 8,094 | 430,415 |
| *B. subtilis* | 3,143 | 6,441 | 5,524 | 3,666 | 77,533 |
| *C. elegans* | 4,098 | 13,915 | 78,152 | 7,849 | 202,109 |
| *D. melanogaster* | 12,823 | 233,054 | 17,530 | 11,774 | 492,331 |
| *D. rerio* | 16,603 | 45,003 | 25,955 | 8,321 | 133,619 |
| *E. coli* | 3,228 | 29,382 | 2,995 | 5,297 | 126,909 |
| *S. cerevisiae* | 7,644 | 164,432 | 237,156 | 8,299 | 328,060 |

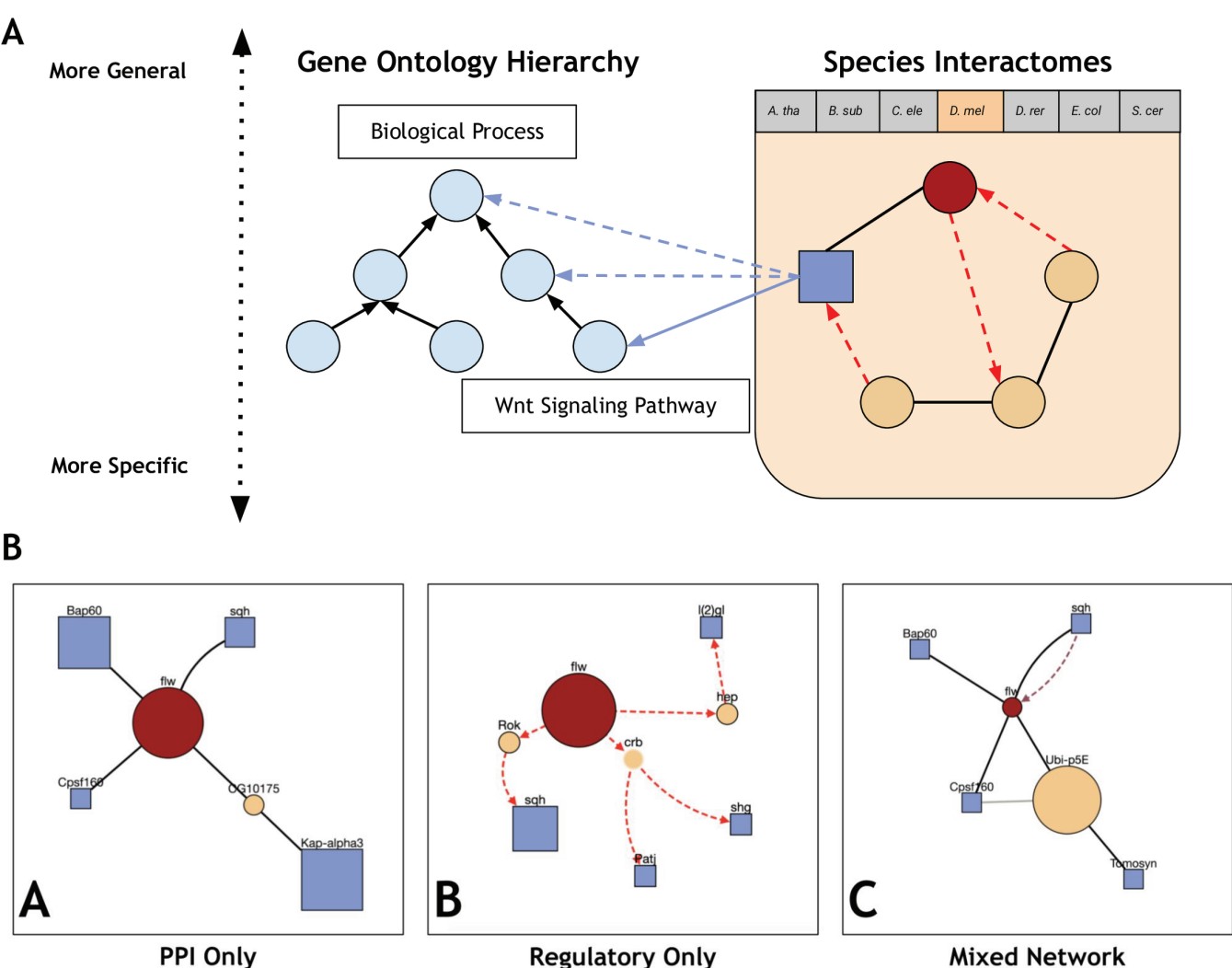

**Fig 1. Network representation and different query examples.** A. Graph representation of Proteins and Gene Ontology terms. Organism-specific networks contain protein-protein interaction edges (black solid lines), regulatory edges (red dashed arrows), and proteins (tan/red circles and blue squares). The GO term hierarchy is represented via the light-blue nodes and directed black solid edges. Proteins directly annotated to GO terms are indicated by blue solid arrows, and blue dashed arrows indicate inferred annotations. B. Different query parameter examples. ProteinWeaver results from a "K Unique Nodes" query connecting flapwing (flw) to $k$=4 nodes annotated to "myosin binding" (GO:0017022) *D. melanogaster* with (B.A) physical interactions, (B.B) regulatory interactions, and (B.C) both physical and regulatory interactions. Dashed directed edges indicate regulatory relationships, solid edges indicate undirected physical relationships; see Fig 4B for a full legend describing node and edge types.

notion of *paths* in the network to visualize nearby connections to a query protein. Paths from $s$ ensure that the subnetwork is connected to the query protein and allows a tunable parameter for the user to view increasingly larger subnetworks. When we visualize the subnetwork, we do not include the GO term $t$, ending the path at the proteins annotated to $t$ (e.g., the square blue nodes in Fig 1B). Users specify the size of the subnetwork returned by selecting a small integer $k$ (which typically ranges between 5 and 25). ProteinWeaver has two different modes for visualizing subnetworks with parameter $k$. In the first mode, "K Unique Paths," Protein-Weaver returns the $k$ shortest paths from $s$ to nodes annotated to $t$ (in terms of the number of edges). K Unique Paths will first visualize immediate neighbors of $s$ that are annotated to $t$, then will visualize nodes further from $s$ as the user increases the parameter $k$. Yen's $k$-shortest

paths algorithm efficiently computes the *k* shortest loopless paths from *s* to nodes annotated to *t* [25]. In the second mode, "K Unique Nodes," ProteinWeaver visualizes the *k* nodes annotated to *t* that are the closest to *s* (in terms of shortest path length). A simple breadth-first search from *s* to reachable nodes annotated to *t* is sufficient for this mode. For both modes, increasing the value of *k* increases the size of the visualized network. More information about the algorithms used for subnetwork generation can be found in Sect A1.5 in S1 File.

ProteinWeaver can return subnetworks consisting of physical, regulatory, or mixed physical and regulatory interactions. The same query run with different interaction types produces different subnetworks (Fig 1B). All three subnetworks included common GO-annotated proteins, such as sqh, but each also identified unique GO-annotated proteins, offering different contexts for the source protein based on interaction type. Researchers can thus choose to query specific networks of interest or combine different interaction types and query modes to explore their protein of interest in a broader biological context.

## Mixed motif enrichment

Many biological processes involve both regulatory and physical interactions. Consequently, representing these networks separately can obscure the complete functional context of a protein of interest [10]. Physical interactions involving TFs have also been shown to effectively predict long-range enhancer-promoter interactions [11]. Therefore, a method for identifying mixed regulatory-physical interaction clusters, or "mixed motifs," is valuable for researchers seeking a comprehensive understanding of the regulation of specific proteins or processes. To provide a view of the patterns associated with a protein or process, we extend a method from Barelvi et al. [26] to calculate motif enrichment statistics for the queried subnetwork.

ProteinWeaver displays statistics for five network motifs that were found to be significantly over-represented in a *S. cerevisiae* mixed PPI-Regulatory network [10]. These 3-node motifs consist of three mixed motifs and two network-specific motifs enriched in the yeast interactome and carry explainable biological significance. The PPI-specific motif, "protein clique" (Fig 2A), often represents three proteins working together in a multi-protein structural or functional unit. The other network-specific motif, "feed-forward loop" (Fig 2B), consists of two TFs, one of which regulates the other, regulating a third protein or gene and is a canonical regulatory motif [27] found to be enriched in *S. cerevisiae*, *E. coli*, and *C. elegans* transcriptional networks [28–30].

In addition to recognizing well-known PPI and gene regulatory motifs, ProteinWeaver can identify mixed network motifs. The first, "interacting coregulators" (Fig 2C), represents two proteins that physically interact and regulate the same gene. The second mixed motif, "coregulated interactors" (Fig 2D), represents two physically interacting proteins that the same

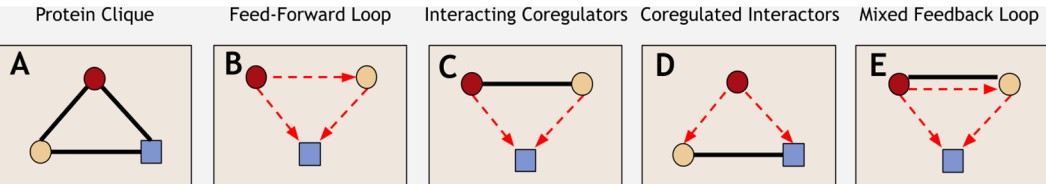

**Fig 2. Mixed motifs in ProteinWeaver.** Mixed network motifs identified by ProteinWeaver; adapted from Yeger-Lotem et al. [10]. Black lines indicate physical interactions. Red arrows indicate regulatory interactions. Motifs can contain multiple protein types.

TF regulates. The final mixed motif, the "mixed feedback loop" (Fig 2E), consists of a feed-forward loop where the two TFs physically interact. Physical interactions between TFs can be regulatory active themselves [11]. Thus, identifying these motifs is valuable for researchers exploring the regulatory implications of physical binding events. Additionally, proteins that work together are often coregulated. Therefore, proteins identified in a coregulated interactor motif may be more likely to have functional impacts on each other [10]. It is important to note that only by using a mixed network can ProteinWeaver capture these last three motifs accurately, as a purely regulatory network would incorrectly classify a mixed feedback loop as a feed-forward loop.

The background distribution of motifs in the *A. thaliana*, *B. subtilis*, *D. melanogaster*, *E. coli*, and *D. rerio* graphs are relatively similar, with all of them having a large number of Protein Cliques relative to the other four motifs (Fig 3). *C. elegans* has a much larger relative proportion of Feed-Forward Loops than the other species and has more evenly distributed motifs than *A. thaliana*, *B. subtilis*, *D. melanogaster*, *E. coli*, and *D. rerio*. *S. cerevisiae* has the most uniformly distributed network, with many instances of all five motifs being found at a relatively high rate. *S. cerevisiae* being the most uniformly distributed may reflect a bias since the motifs identified here are based on motifs found enriched in a yeast network [10]. For a queried subnetwork, ProteinWeaver counts the number of each type of motif in the network (Fig 4G). Enrichment scores, $Z$-scores, and $p$-values are calculated compared to the species-specific background graph; see Sect A2 in S1 File for more information.

## Network approaches for GO term annotation prediction

In a typical ProteinWeaver query, the query protein *s* is not associated with the queried GO term *t*, and the goal is to connect the query *s* to proteins annotated to *t*. Sometimes, the queried protein *should* be considered part of the GO term, but there is not yet evidence in the Gene Ontology to properly annotate the protein. To provide the user additional context about whether the query is likely to be associated with the GO term of interest, we used a random walk approach to assign a confidence score about whether a query protein *s* is "near" GO-term annotated proteins. The random walk scoring function runs personalized PageRank [31], restarting from proteins annotated to the GO term *t* with a damping factor $\alpha = 0.7$. We rank the query node *s* according to the final visitation probabilities of the random walk; see Sect A3 in S1 File for more details.

We compare the random walk approach (which we call **RandomWalk**) to three other prediction scores based on neighbor overlaps in the graph:

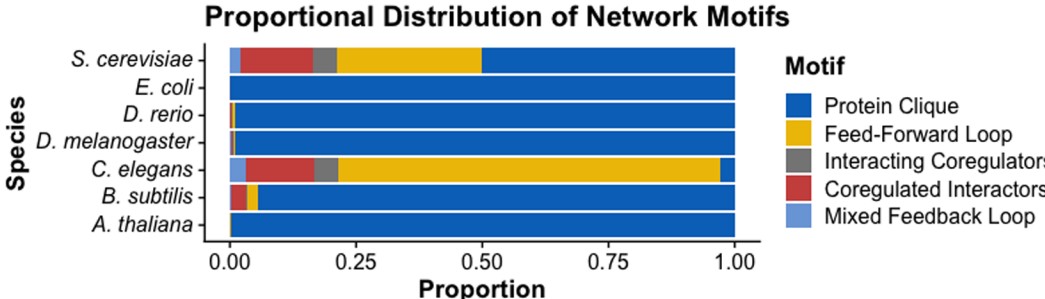

**Fig 3. Proportional distribution of network motifs.** For a detailed breakdown of the motifs by species, see Sect A2.2 in S1 File.

**Degree:** Rank $s$ by its degree in the original graph $G$ (larger is better). This approach does not account for the GO term $t$, and it is used to assess degree bias in our evaluation.

**One-Hop GO Overlap:** Rank $s$ by the number of $s$'s neighbors that are annotated to GO term $t$ (larger is better).

**Hypergeometric Distribution:** Rank $s$ by the hypergeometric distribution $p$-value of observing the number of $s$'s neighbors annotated to GO term $t$, adjusted by the size of the GO term $t$ and the degree of $s$ (smaller is better).

See Sect A3.1 in S1 File for more details about these comparator methods.

## Results

We first describe the ProteinWeaver interface and illustrate the potential of ProteinWeaver in generating hypotheses with two case studies from the recent literature, then highlight the protein function prediction features of the tool.

### ProteinWeaver interface

ProteinWeaver allows users to explore networks for non-human model organisms by inputting a protein, GO term, and a size parameter ($k$), specifying whether to traverse a PPI network, GRN, or a mix of both (Figure 4A). The induced subnetwork gives users a detailed view of how nodes are interconnected within the organism (Fig 4B). Users can navigate the network by selecting GO terms from the hierarchy or changing the queried protein to any protein within the subnetwork, fostering hypothesis generation based on the species' interactome (Fig 4D). For each query, ProteinWeaver provides comprehensive details, including annotation links for the queried protein and GO term, GO term definitions, and links to organism-specific databases (Figs 4C & 4E). The tool also includes a statistics section showing graph statistics, GO term annotation confidence scores, and mixed motif data (Figs 4C, 4E, & 4G). To support reproducibility, queries can be saved as hyperlinks or exported as PNGs, JSON files, or Cytoscape objects for further analysis.

### Eb1's role in microtubule bundle formation in *D. melanogaster*

The protein Eb1 is part of a group of the end-binding proteins family responsible for microtubule plus end growth [32]. Microtubules are important polymers in all eukaryotes that play the role of changing a cell's shape, division, and transport [33]. Microtubules interact with microtubule-associated proteins (MAPs) to regulate microtubules in the cell [33]. In a 2021 paper, the gene Eb1, along with Tau and XMAP215/Msps were discovered to cooperate independently in the axon of *Drosophila* to regulate microtubule polymerization and bundle formation [34]. Furthermore, in a 2013 paper, Eb1 was noted to be important in apicobasal microtubule bundle formation and epithelial elongation [35]. However, in the Gene Ontology database, Eb1 has not yet been annotated to microtubule bundle formation in *Drosophila*.

We queried the connections between Eb1 and microtubule bundle formation and generated a visualization of Eb1's connection to other microtubule bundle-related proteins (Fig 5). The queries are also available at https://bit.ly/fly-paths-mode and https://bit.ly/fly-nodes-mode. The two modes, "K Unique Paths" and "K Unique Nodes," produced subnetworks with different network topologies: both modes used $k$ = 15, but 15 paths reached 10 nodes annotated to microtubule bundle formation (Fig 5A). On the other hand, a network that reached 15 GO term-associated nodes resulted in a larger network with seven intermediary proteins (Fig 5B). Note that the intermediate nodes from Fig 5A (Stat92E and Su(dx)) also

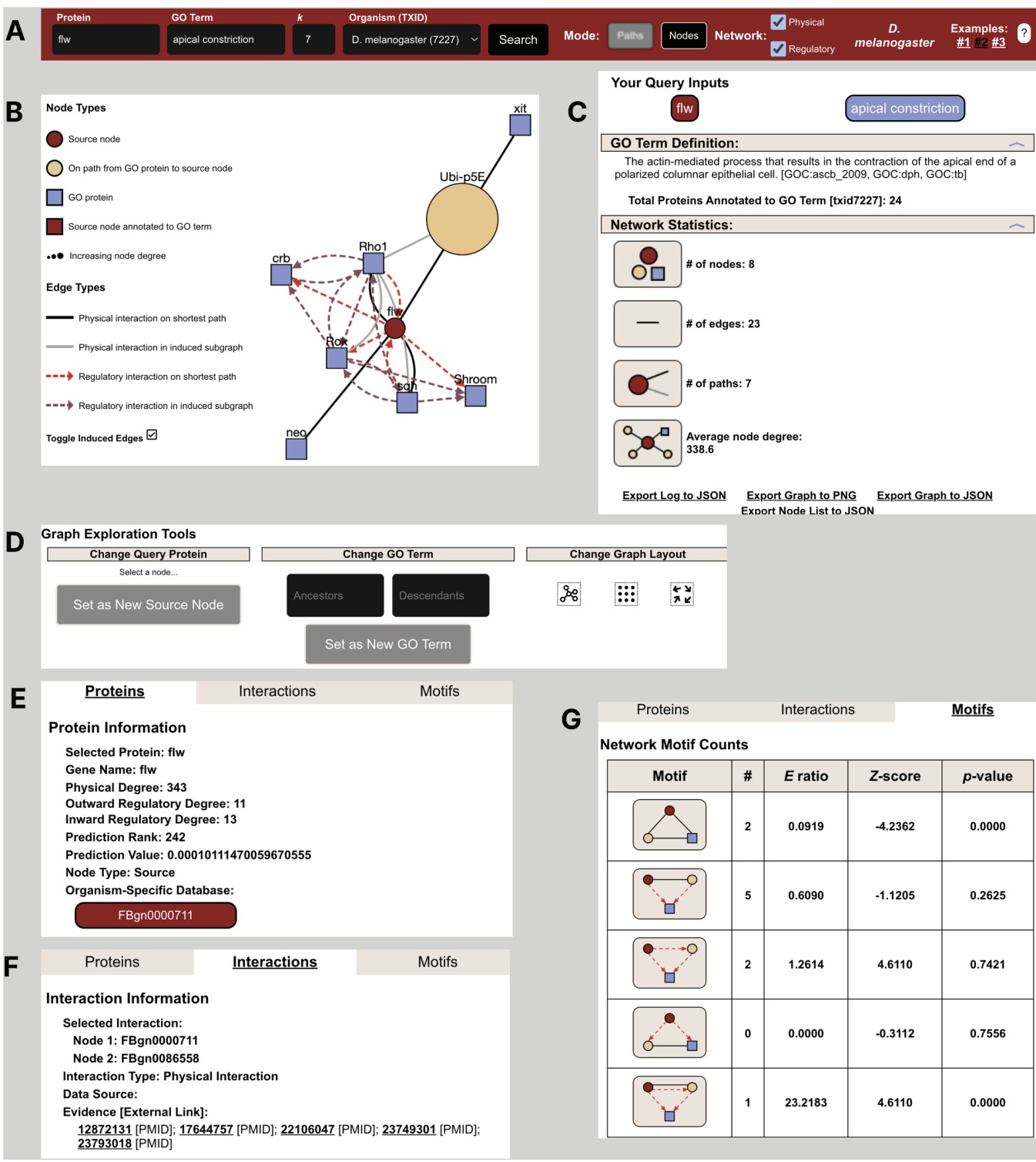

**Fig 4. ProteinWeaver interface** A) Users enter their protein and GO term of interest, a small integer *k*, and the species in the top panel and click "Search". B) The subnetwork is shown via `react-cytoscapejs`'s plugin. The legend section details the different node and edge types. C) The sidebar includes links to protein and GO term data, basic network statistics, and the ability to export the session. D) The graph exploration section lets users rearrange the network layout and update the query by selecting a new query node or traversing the gene ontology. E-G) The tabbed window shows information about the user-selected protein (including the protein's rank relative to the GO term), information about the user-selected interaction, and the motifs found in the subnetwork.

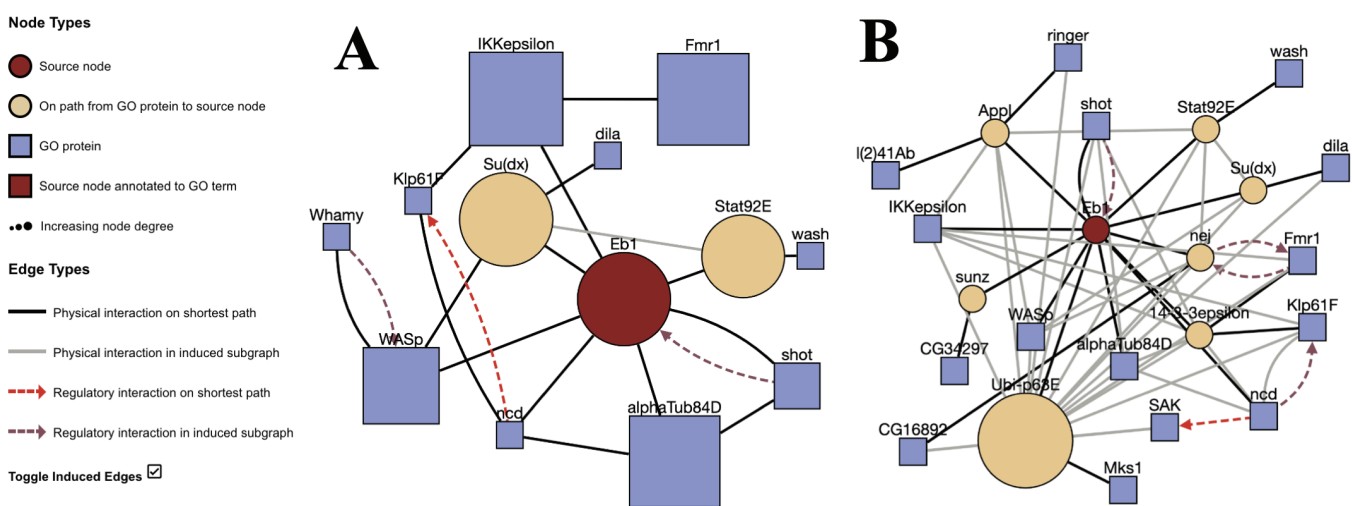

**Fig 5. Eb1 to "microtubule bundle formation" (GO:0001578) in *D. melanogaster*.** (A) "K Unique Paths" mode and (B) "K Unique Nodes" mode with $k = 15$.

appeared in Fig 5B, but their node size was dwarfed by the large number of Ubi-p63E's neighbors (Ubi-p63E has 5,066 physical interactions). Ubi-p63E's role in ubiquitination affects the vast majority of biological processes in the cell, so it was not surprising that it appeared with many connections in the queried subnetwork.

In both visualizations, Eb1 was connected to important microtubule-associated proteins such as Short stop (Shot), known to bind actin and microtubules, and Ncd, a minus-end-directed kinesin microtubule motor protein [36,37]. ProteinWeaver also identified two Wiskott-Aldrich Syndrome (WAS) family proteins, Wash and WASp, which were directly annotated to microtubule bundle formation. WASp and Wash directly bundle microtubules and are essential for oocyte formation. Wash has also been shown to crosslink F-actin and microtubules together [38]. Additionally, ProteinWeaver identified the proteins Su(dx) and Stat92E in both path-finding modes. These proteins lacked direct annotation to microtubule bundle formation, but linked the query protein to microtubule-associated proteins. Su(dx) is important during oogenesis, where it regulates the formation of interfollicular stalks, the separation of the egg chamber, and the enwrapment of germline cysts by follicular stem cells [39]. Stat92E has been shown to be essential in various developmental processes, particularly those involving morphological changes or cell movement during *Drosophila* embryogenesis [40].

## Gdf6a's role in dorsal/ventral patterning in *D. rerio*

Bone morphogenic proteins (BMPs) have been shown to function in a variety of processes in animals, including patterning and differentiation of tissues, establishing cell polarity, maintaining organ homeostasis, and responding to injuries [41]. The BMP pathway in zebrafish is regulated in part by the Smad family of transcription factors [42]. Further, BMPs have been found to regulate Smads through phosphorylation via a receptor complex [41]. Previous work has shown that knocking out a specific BMP, *gdf6a*, blocks retinal Smad phosphorylation, indicating that Gdf6a is involved in regulating Smad proteins in the retina [43].

We visualized the connection between Gdf6a and other proteins linked to dorsal/ventral (D/V) patterning to investigate how Smad transcription factors might integrate bone morphogenic proteins with other signaling pathways to promote D/V pattern formation (Fig 6). The queries are also available at https://bit.ly/zebrafish-paths-mode and https://bit.ly/zebrafish-nodes-mode. In this example, Gdf6a was annotated to D/V pattern formation (denoted by the red square in Fig 6); ProteinWeaver was used to find additional connections from Gdf6a to other members of the same GO term. Again, the mode of visualizing $k = 15$ unique paths reached fewer nodes annotated to D/V patterning than the other mode, which visualized the connections needed to reach $k = 15$ nodes annotated to D/V patterning.

In both visualizations, ProteinWeaver identified a receptor protein serine/threonine kinase ACVR1 that connected Gdf6a to proteins annotated to dorsal/ventral pattern formation (Fig 6). ACVR1 interacted with Gdf6a, Bmp7a, Bmp2B, and Smad5 and is involved in left/right patterning in mice development [44]. These proteins interacted with more Smad family transcription factors, which was shown especially in the K Unique Nodes mode (Fig 6B). Smad2 was particularly interesting because it was not annotated to the queried GO term "dorsal/ventral pattern formation," and Smad 1/5/9 are usually associated with this patterning in zebrafish [45]. However, Smad2 has been linked to dorsal mesoderm specification in *Xenopus*, and mutants have shown that Smad2 is essential in early embryo patterning events in mice [46].

## Predicting missing GO term annotations

We first assessed the four proposed methods for ranking protein-GO associations based on the underlying network (One-Hop GO Overlap, Hypergeometric Distribution, Degree, and RandomWalk; see Methods). For each species, we generated a dataset of 1,000 positive protein-GO pairs by randomly choosing an existing protein-GO edge and modifying the graph $G$ to remove that specific edge. We expected these nodes to have high confidence in their membership with the GO term. For every positive protein-GO pair, we selected 100 negative proteins by identifying proteins that (1) were near the positive protein when considering only the PPI and regulatory edges, (2) were not connected to the positive GO term and

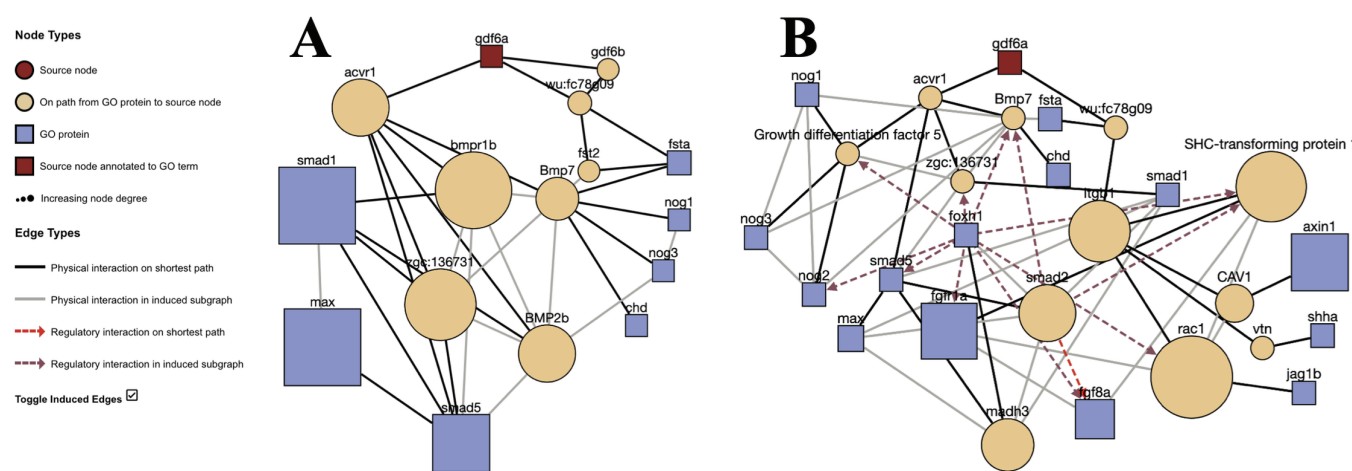

**Fig 6. Gdf6a to "dorsal/ventral pattern formation" (GO:0009953) in *D. rerio*.** (A) "K Unique Paths" mode and (B) "K Unique Nodes" mode with $k = 15$.

(3) had approximately the same degree as the positive node. See Sect A3.2 in S1 File for more information about positive and negative sampling and the evaluation pipeline.

For each species, we plotted Receiver Operator Characteristic (ROC) and precision-recall curves (Fig 7 and Fig 8). The RandomWalk method (in orange) dominated in both metrics across all species: RandomWalk had a nearly perfect ROC AUC across all species and also

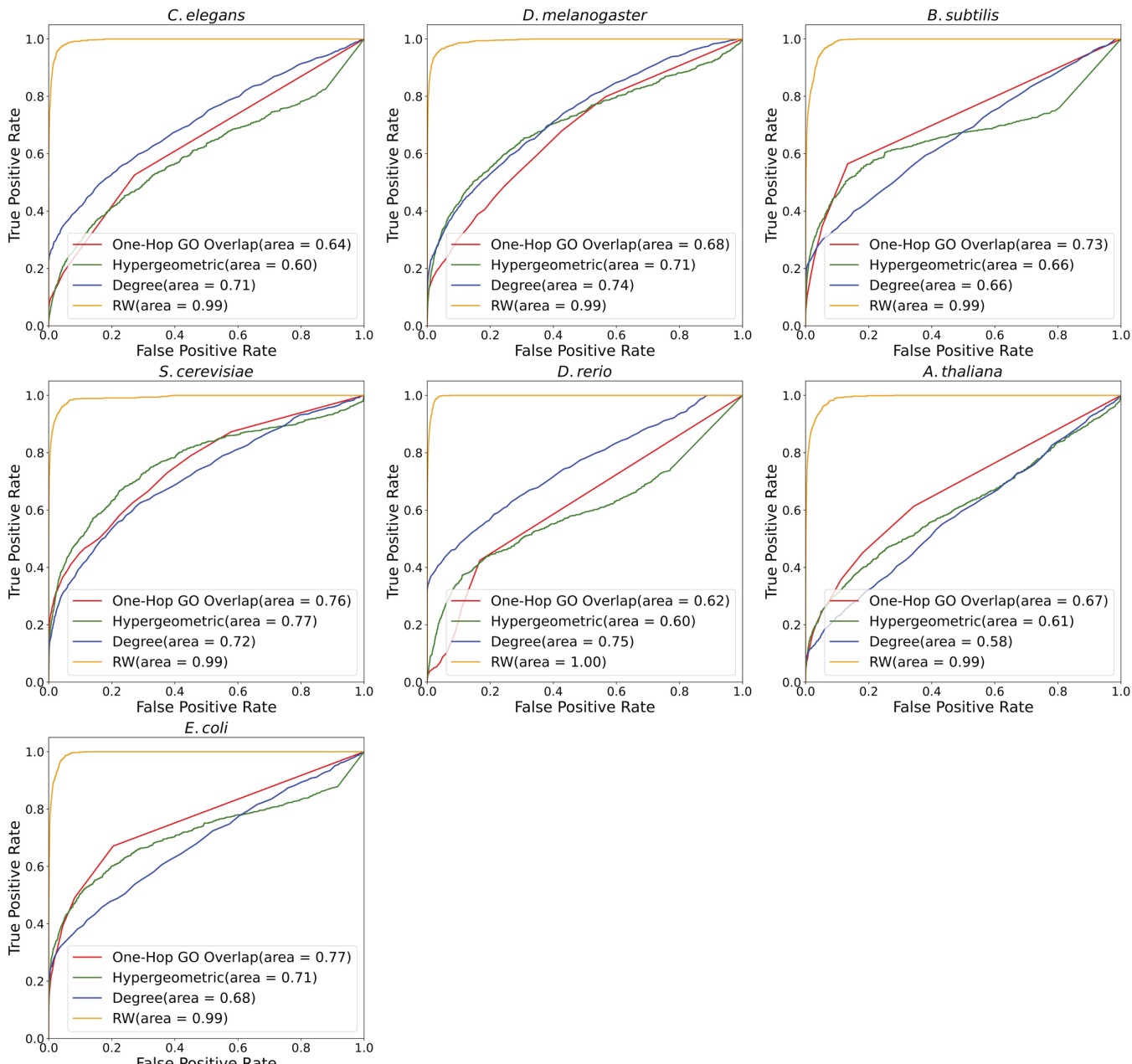

**Fig 7. ROC curves of four annotation prediction methods.** Receiver Operator Characteristic (ROC) curves of the One-Hop GO Overlap, Hypergeometric Distribution, Degree, and RandomWalk annotation prediction methods for all species. The ROC AUC is reported in the legend.

Precision/Recall Curve for All Species w/ Complete Inferred Networks

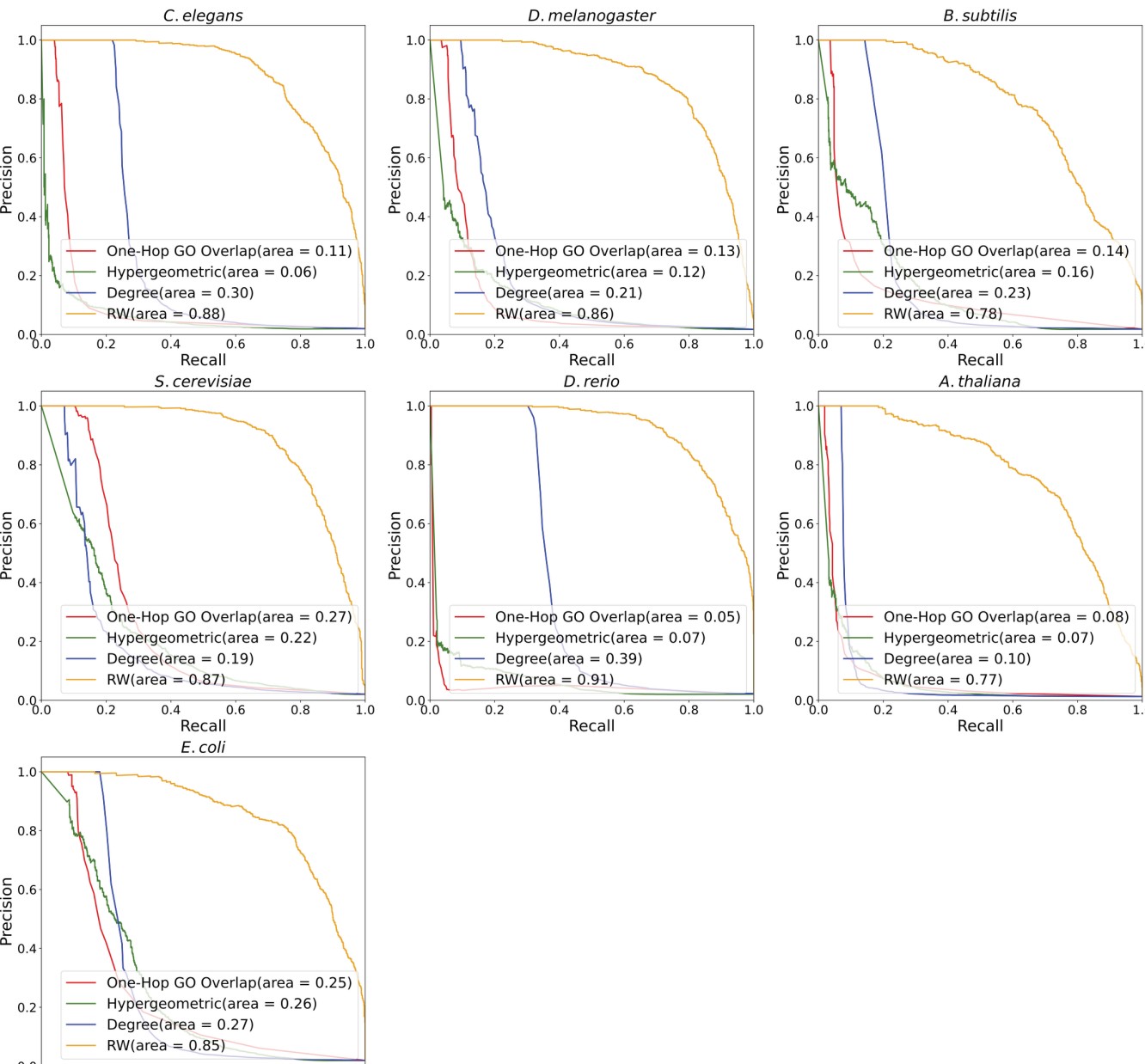

**Fig 8. Precision-Recall curves of four annotation prediction methods.** Precision-Recall curves of the One-Hop GO Overlap, Hypergeometric Distribution, Degree, and RandomWalk annotation prediction methods for all species. The PR AUC is reported in the legend.

had the highest precision at larger recall values. The rest of the methods performed considerably worse than RandomWalk. After RandomWalk's ROC AUC of 0.99, the next best ROC AUC was 0.77 (e.g., the Hypergeometric method in *S. cerevisiae* and the One-Hop GO Overlap in *E. coli*, Fig 7). The story was similar for PR AUC: RandomWalk's worst PR AUC was 0.77, whereas the next best PR AUC was 0.39 (e.g, the Degree method in *D. rerio*). The

Degree method seemed to be better than the other two for *C. elegans* and *D. rerio*, suggesting a degree bias in these species. Additionally, the One-Hop GO Overlap method seemed to be better than the other two for *S. cerevisiae*. However, the performance of all three methods was poorer compared to the RandomWalk approach.

The background network included directly annotated protein-GO term pairs as well as inferred annotations between proteins and more general GO terms (see Methods). One reason the RandomWalk might work so well is that it could rely on the inferred protein-GO term pairs rather than the direct annotations. To check this, we also ran the four protein function prediction methods on the graphs with only directly-annotated networks; while the relative ordering of the three comparator methods changed, the random walk approach remained superior (Sect A3.3 in S1 File). Furthermore, we tested the performance of RandomWalk on background networks that only consisted of protein-GO term annotations (removing the protein interactions) and found that RandomWalk performed only slightly worse in terms of ROC and precision-recall (Sect A3.4 in S1 File).

The RandomWalk protein function prediction feature provides the users an intuitive notion about connectedness between a protein and the GO term of interest. For example, the RandomWalk rankings provided additional context to the two case studies above. In *D. melanogaster*, Eb1 was ranked 86th out of the 12,800 fly proteins not annotated to microtubule bundle formation. In comparison, RandomWalk ranked the ubiquitinase Ubi-p63E 4,786th, reflecting the many connections Ubi-p63E has in the network. In *D. rerio*, the query Gdf6a was already part of the "dorsal/ventral pattern formation" GO term, so it was not surprising to be ranked 5th out of the 16,000 nodes not annotated to D/V patterning. However, ProteinWeaver ranked Smad2 53rd, which was still quite high. Surprisingly, ACVR1 was ranked 4,013th: in this case, the RandomWalk ranked ACVR1 lower down the list because there were only 15 physical interactions with other nodes (making it less connected to D/V patterning nodes than other proteins).

## Discussion

ProteinWeaver is designed to be an efficient, user-friendly, and reproducible tool for GRN and PPI network analysis. ProteinWeaver provides researchers with novel network querying capabilities of protein interactions, regulatory interactions, or a combination of interaction types for seven species. Mixed motif identification and GO term annotation predictions serve as additional resources for hypothesis generation and network exploration in these non-human model organisms.

We note that our approach to predict GO term annotations is a different problem than that of determining protein function, which is a longstanding challenge that has seen explosive improvements with structural alignments provided by deep learning models such as AlphaFold [47]. Here, we aim to use the physical, regulatory, and GO-annotated relationships to assess how "near" a protein is to a GO term without additional information such as protein sequences, domains, or structure.

Currently, ProteinWeaver supports seven non-human model organisms but plans to expand to include more, e.g., *Xenopus laevis*, *Zea mays*, and *Pisum sativum*. Current limitations are related to the lack of freely available physical and regulatory network data for specific species. We are also working to provide more context about the local subnetwork structure in relation to the GO-annotated proteins and other nearby interacting molecules beyond the current basic statistic. This context will come in the form of expanded network statistics, such as centralities and clustering coefficients, modified to account for proteins annotated to the GO term of interest.

In addition, ProteinWeaver aims to improve its querying capabilities by allowing users to search with multiple proteins of interest or multiple GO terms of interest. This facilitates network exploration for researchers interested in how several proteins interact within a specific biological process or how multiple biological processes may connect.

ProteinWeaver aims to maintain an intuitive and straightforward user interface while providing dense information. Striking the right balance between these goals poses a challenge, and further enhancements are made continually to optimize user experience while maintaining informativeness. ProteinWeaver invites collaboration with the scientific community for feature development. Open discussions and feedback sessions will guide the implementation of features aligned with user needs and advancements in biological research.

In conclusion, ProteinWeaver facilitates the understanding of molecular interactions and their roles in biological contexts, even for those without computational expertise. Its intuitive interface and GO term integration address challenges researchers face in situating proteins within a specific biological context. ProteinWeaver is positioned to be a useful tool for hypothesis generation and biological interaction exploration for researchers studying non-human model organisms.

## Supporting information

**S1 File. Supplementary text, figures, and tables.**
(PDF)

## Acknowledgments

We thank Larry Zeng for his help with the web server and the tech stack architecture. We also thank Derek Applewhite, Kara Cerveny, and Shivani Ahuja for their collaborative discussions.

## Author contributions

**Conceptualization:** Oliver Anderson, Altaf Barelvi, Anna Ritz.

**Data curation:** Oliver Anderson, Altaf Barelvi.

**Funding acquisition:** Anna Ritz.

**Investigation:** Oliver Anderson, Altaf Barelvi, Anna Ritz.

**Methodology:** Oliver Anderson, Altaf Barelvi, Aden O'Brien, Ainsley Norman, Iris Jan, Anna Ritz.

**Project administration:** Anna Ritz.

**Software:** Oliver Anderson, Altaf Barelvi, Aden O'Brien, Ainsley Norman, Iris Jan, Anna Ritz.

**Supervision:** Anna Ritz.

**Validation:** Oliver Anderson, Altaf Barelvi, Anna Ritz.

**Visualization:** Oliver Anderson, Altaf Barelvi, Anna Ritz.

**Writing – original draft:** Oliver Anderson, Altaf Barelvi, Anna Ritz.

**Writing – review & editing:** Oliver Anderson, Altaf Barelvi, Aden O'Brien, Ainsley Norman, Iris Jan, Anna Ritz.

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
