## [Decision Letter · Decision Letter 0]

17 Jun 2025

PONE-D-24-52122ProteinWeaver: A Webtool to Visualize Ontology-Annotated Protein NetworksPLOS ONE

Dear Dr. Ritz,

Thank you for submitting your manuscript to PLOS ONE. After careful consideration, we feel that it has merit but does not fully meet PLOS ONE’s publication criteria as it currently stands. Therefore, we invite you to submit a revised version of the manuscript that addresses the points raised during the review process.

We look forward to receiving your revised manuscript.

Kind regards,

Abozar Ghorbani, Ph.D

Academic Editor

PLOS ONE

Journal Requirements:

2. Please expand the acronym “DBI” (as indicated in your financial disclosure) so that it states the name of your funders in full.

We thank Larry Zeng for his help with the web server and the tech stack architecture. We also thank Derek Applewhite, Kara Cerveny, and Shivani Ahuja for their collaborative discussions. This work was funded by NSF-DBI-1750981 (awarded to AR).

This work was supported by a National Science Foundation (https://www.nsf.gov/) grant NSF-DBI-1750981 awarded to AR. The funders not play any role in the study design, data collection and analysis, decision to publish, or preparation of the manuscript.

4. Thank you for uploading your study's underlying data set. Unfortunately, the repository you have noted in your Data Availability statement does not qualify as an acceptable data repository according to PLOS's standards.

Reviewers' comments:

Reviewer's Responses to Questions

**Comments to the Author**

1. Is the manuscript technically sound, and do the data support the conclusions?

Reviewer #1: Yes

Reviewer #2: Yes

2. Has the statistical analysis been performed appropriately and rigorously? 

Reviewer #1: N/A

Reviewer #2: Yes

3. Have the authors made all data underlying the findings in their manuscript fully available?

Reviewer #1: Yes

Reviewer #2: Yes

4. Is the manuscript presented in an intelligible fashion and written in standard English?

Reviewer #1: Yes

Reviewer #2: Yes

5. Review Comments to the Author

Reviewer #1: 1- Please clarify the reasons of choosing Yen’s and breadth-first search algorithms.

2- It is better to discuss the used algorithms and their potentials more in the results and discussion.

3- In this 5-model organisms based tool, E. coli is totally neglected. Although the authors brought the topic in Discussion section and trying to justify it, the reasons of this negligence were not enough.

Reviewer #2: Hello dear authors

Based on the manuscript, the following comments should be resolved before the final editor decision on the manuscript.

1. It is highly recommended to make a logical relationship between the following paragraph of “Introduction” section and the previous one, and highlight the importance of your study.

“We present ProteinWeaver, a molecular interaction network visualization tool that generates subnetworks of physical and regulatory interactions based on a protein and a biological function of interest for non-human model organisms. In contrast to previous tools, ProteinWeaver links proteins to relevant biological processes, provides customizable network visualizations, and encourages interactive network exploration. To our knowledge,”

2. With reference to this sentence of “Introduction” section: “Currently, ProteinWeaver supports a prokaryote (the Gram-positive bacterium Bacillus subtilis subsp. subtilis str. 168), a single-celled eukaryote (the brewer’s yeast Saccharomyces cerevisiae S288C), two morphologically-distinct invertebrates (the fruit fly Drosophila melanogaster and the nematode Caenorhabditis elegans), and a vertebrate (the zebrafish Danio rerio)….” : It is very informative to select a representative from various organisms. However, it is highly encouraged to include a representative of viruses such as SARS-CoV-2 to make a comprehensive server that allow this valuable analysis workflow to users.

3. Some errors exist in the manuscript such as “single-celled” which should be replaced by “unicellular”. Moreover, all scientific names should be written in italics.

4. The Saccharomyces cerevisiae is considered as the unicellular eukaryote, but the yeasts are mainly considered as a prokaryote.

5. Table captions should be inserted above the Tables. Moreover, please cite this Table below the related text. It is not common to start the manuscript section with a table or figure.

6. Some grammar issues and vague sentences exist. So, it is highly recommended to revise the manuscript accordingly and resolve this issue.

7. Some repetitive structures exist throughout the manuscript such as the following one in “Materials and Methods” section: "Currently, ProteinWeaver supports a prokaryote (the Gram-positive bacterium Bacillus subtilis subsp. subtilis str. 168), etc...." The authors are highly encouraged to place this information here and remove it from introduction.

8. In “2.2.” section, Figure 3.A should be written as Figure 2.

9. It is not common to state both full structure and abbreviation of each term after the first definition. Please revise the manuscript.

10. It is not common to discuss the literatures in results section, unless results and discussion being mixed.

11. It is highly recommended to revise lengthy sentences like these: “We assessed the four methods by generating a dataset of 1,000 positives, which were selected by randomly choosing a protein-GO term edge and modifying G to remove that specific edge. We expect these nodes to have high confidence in their membership with the GO term. For every positive (protein-GO Term pair), we selected 100 negative proteins by identifying proteins that are (1) near the positive protein when considering PPI and regulatory edges only, (2) are not connected to the positive GO term, and (2) have approximately the same degree as the positive node.”. Such structures could confuse readers. Please revise the manuscript and resolve this issue accordingly.

12. It might that large explanations like this, with the nested examples confuse the readers and decrease the value of the server usage. It is highly recommended to revise the manuscript and resolve the similar structures like this: “We found that RandomWalk had a nearly perfect ROC AUC across all species (Figure 6. RandomWalk also had the highest precision at varying recall values and it drops off precision at higher recall values. The rest of the methods performed considerably worse than RandomWalk in the ROC AUC analysis. Between the methods, their ranks varied in different species analysis for both the inferred and non-inferred networks. For example, the degree method was ranked second in the ROC value for C. elegans, however was ranked 4th for the yeast dataset. The Precision/Recall values for the One-Hop GO Overlap and Degree methods noticed an increase when using the inferred networks. The significance of this increase varied among the methods and which species. For example, the degree method noticed the largest Precision/Recall increase in the C. elegans dataset. The Hypergeometric method, however, did not show any noticeable difference between the Precision/Recall values across all the species when using the inferred and non-inferred networks.”

13. The manuscript has many vague structures like “This could be because adding inferred protein-GO edges did not affect the overall hypergeometric equation used to calculate their scores. We also ran these methods on the graphs with only directly-annotated networks; while the relative ordering of the three comparator methods changed, the random walk approach remained superior”. Please revise whole document and correct it.

14. It is better to mention case studies in materials and methods section and only state the results here.

15. It is not necessary to redundant explanations such as the microtubule or BMPs importance.

16. All phrases about the gathered results should be write in past tense.

17. The authors have been provided a valuable web-server. However, it is recommended to improve the graphical representation.

Sincerely

6. PLOS authors have the option to publish the peer review history of their article (what does this mean?). If published, this will include your full peer review and any attached files.

Reviewer #1: No

Reviewer #2: No

---

## [Author Response · Author response to Decision Letter 1]

25 Jul 2025

See the Response to Reviewers PDF uploaded with the manuscript files.

---

## [Decision Letter · Decision Letter 1]

14 Aug 2025

ProteinWeaver: A webtool to visualize ontology-annotated protein networks

PONE-D-24-52122R1

Dear Dr. Ritz,

We’re pleased to inform you that your manuscript has been judged scientifically suitable for publication and will be formally accepted for publication once it meets all outstanding technical requirements.

Kind regards,

Abozar Ghorbani, Ph.D

Academic Editor

PLOS ONE

Additional Editor Comments (optional):

Reviewers' comments:

Reviewer's Responses to Questions

**Comments to the Author**

1. If the authors have adequately addressed your comments raised in a previous round of review and you feel that this manuscript is now acceptable for publication, you may indicate that here to bypass the “Comments to the Author” section, enter your conflict of interest statement in the “Confidential to Editor” section, and submit your "Accept" recommendation.

Reviewer #1: (No Response)

Reviewer #2: (No Response)

2. Is the manuscript technically sound, and do the data support the conclusions?

Reviewer #1: Yes

Reviewer #2: Yes

3. Has the statistical analysis been performed appropriately and rigorously? 

Reviewer #1: Yes

Reviewer #2: N/A

4. Have the authors made all data underlying the findings in their manuscript fully available?

Reviewer #1: Yes

Reviewer #2: Yes

5. Is the manuscript presented in an intelligible fashion and written in standard English?

Reviewer #1: Yes

Reviewer #2: Yes

6. Review Comments to the Author

Reviewer #1: (No Response)

Reviewer #2: Hello dear authors

Thanks for the valuable improvement of the manuscript. Please resolve the following comments before the final editor decision on the manuscript.

1. As it was pointed out in the previous review, it is not common to state both full structure and abbreviation of each term after the first definition. The first use of PPI, GRN, TF terms is in the "introduction section". There is an inconsistency throughout the manuscript, accordingly. The authors use the full structure and abbreviations following each other. Please revise the manuscript.

2. As it was pointed out in the previous version, all scientific names should be given in italics (e.g., line 224). Please revise whole manuscript and resolve this issue.

3. All phrases about the gathered results should be write in past tense.

Sincerely

7. PLOS authors have the option to publish the peer review history of their article (what does this mean?). If published, this will include your full peer review and any attached files.

Reviewer #1: **Yes: **Nastaran Asghari Moghaddam

Reviewer #2: No

---

## [Editor Report · Acceptance letter]

PONE-D-24-52122R1

PLOS ONE

Dear Dr. Ritz,

I'm pleased to inform you that your manuscript has been deemed suitable for publication in PLOS ONE. Congratulations! Your manuscript is now being handed over to our production team.

Kind regards,

on behalf of

Dr. Abozar Ghorbani

Academic Editor

PLOS ONE